refugee; migrant; victimized child; achievement emotion

**Corresponding author:**
Engin Karadag;
Email: engin.karadag@hotmail.com

# Achievement emotion in war victim children: A study on Syrian primary and secondary school students in Turkey

Esra Sarac[1], Fatih Bektas[1], Emine At[2], Engin Karadag[2,3] and S. Koza Ciftci[2,3]

[1]Kilis 7 Aralik University, Kilis, Türkiye; [2]Akdeniz University, Antalya, Türkiye and [3]Khazar University, Baku, Azerbaijan

## Abstract

The Syrian government's violent suppression of pro-democracy protests in March 2011 sparked a civil war that resulted in the deaths of hundreds of thousands of people and the displacement of millions. This study focuses on the emotional achievement of 357 Syrian primary and secondary school students who have moved to Türkiye and are under temporary protection. The researchers used the achievement emotion scale to collect data. They conducted a *t*-test, analysis of variance, correlation analysis and multiple linear regression to examine the sociodemographic factors affecting students' achievement emotions. The results revealed that boy students experienced more negative achievement emotions than girl students, and that the longer the students have been in temporary protection, the more their positive achievement emotions have decreased. The ongoing war in Syria has dire consequences for school-age children who have been forced to flee their homes.

## Impact statement

This study provides new insights into the achievement emotions of Syrian refugee students living in Türkiye. Achievement emotions, such as enjoyment, pride, boredom and anxiety, play a critical role in students' motivation, learning and engagement. However, little is known about how these emotions develop among refugee children who face challenges such as language barriers, cultural adaptation and uncertainty about their future. The findings reveal that refugee students generally experience more positive than negative achievement emotions (NAEs), which vary depending on individual and contextual factors. For example, girls tend to report stronger positive feelings of achievement, while longer residence in the host country is linked with higher NAEs. These results highlight the complex patterns of achievement emotions that refugee children navigate as they adapt to a new culture and schooling environment. In a broader sense, this research extends the concept of achievement emotions beyond traditional student populations and cultural settings. It underscores the importance of considering achievement emotions in educational policies and practices, particularly in refugee education contexts. By doing so, the study offers insights that may guide interventions to support students' achievement emotions and overall engagement in learning.

## Introduction

Many countries across the globe have experienced a surge in immigrant populations, including student populations. Europe, for instance, has faced the daunting challenge of incorporating substantial numbers of newly arrived migrants and refugees into its educational system over the past decade. The United States currently holds the distinction of harboring the highest number of immigrants worldwide, and the children of these immigrants must be enrolled in school systems (Martin et al., 2024).

The refugee crisis has impacted Türkiye as a result of the Syrian civil war, which began in 2011. Due to ongoing conflict, numerous individuals have been displaced by cross-border migration. According to official data, Türkiye hosts the highest number of Syrian asylum seekers worldwide, with 3,559,041 Syrians under temporary protection (Republic of Türkiye's presidency of migration management). Of these immigrants, 1,124,353 were school-age children. The Turkish Ministry of National Education guarantees these children's rights to basic education. Since 2011, Türkiye has endeavored to provide educational opportunities for Syrian children. However, the situation has presented numerous challenges for education stakeholders who have not encountered many migrant students (UNICEF, 2022). This rapid and intense wave of migration has had and continues to have various effects on the fields of culture, society, politics and economics in Türkiye (Tuncel and Ekici, 2019). Education began in refugee camps between 2011

and 2013 and has since continued with the integration of Syrian students into public schools. However, some children do not have access to education. It was observed that some children do not participate in education at all, and a significant portion of them drop out of school, particularly in high school. As of November 2021, only 65% of Syrian children attended school (UNICEF, 2022).

Syrian students who attend school encounter a multitude of issues. Among these, language barriers, differences in the education system and cultural differences stand out as the most significant challenges faced by students in schools (Çelik et al., 2021). These issues have both academic and psychological implications. In addition to economic and cultural circumstances, some school dropouts may be attributed to a lack of motivation, bullying or isolation (UNICEF, 2022). Consequently, it is crucial to promote the integration of immigrant students into schools. If these students fall behind academically, they may be at risk of not acquiring the critical skills necessary for achievement, including after-school education, training and employment (OECD, 2006). Therefore, the "achievement emotion," which is believed to influence the academic performance and adjustment of immigrant students in school, is the focus of this study.

According to Pekrun and Linnenbrink-Garcia (2014), emotions are experienced in educational settings and serve to achieve academic achievements and personal growth. Emotions and motivation are essential components of learning and achievement processes, and they act as prerequisites, agents and outcomes of these experiences (Schukajlow et al., 2017). Furthermore, emotions can significantly impact motivation, cognitive resource activation, learning behavior and student assessment results (Pekrun, 2016).

In the school environment, students experience various emotions (Pekrun and Linnenbrink-Garcia, 2014). These emotions are intertwined with the learning and teaching conditions, cognitive knowledge production process and social interactions that occur in the classroom (Pekrun et al., 2005; Frenzel et al., 2007; Pekrun and Linnenbrink-Garcia, 2014). Therefore, it is crucial to comprehend the nature of emotions within the school context, as they are intimately linked to almost all aspects of teaching and learning processes (Schutz and Lanehart, 2002).

Previous investigations have primarily focused on the emotions resulting from achievements and failure, while neglecting the emotions connected to the activity (Pekrun and Stephens, 2010). In contrast, achievement emotions provide a more comprehensive perspective, addressing both activity- and outcome-related emotions. Achievement emotions are directly linked to achievementful activities or outcomes (Pekrun, 2006). Additionally, achievement emotions are interconnected through multifaceted processes (Pekrun et al., 2023). For example, one may feel bored in class, anxious when facing challenging tasks or experience joy upon completing a difficult task.

Furthermore, hopelessness may arise when one believes they will not receive a good grade on an exam. It is important to note that contextual factors can significantly affect individuals' motivation and emotions in situations involving achievement. Consequently, achievement emotions can be experienced in various achievement-focused settings, such as classrooms, exams and homework (Putwain et al., 2022).

The emotion of achievement is influenced by gender, the social environment and cultural context (Pekrun, 2018). Control-value theory posits that the sociocultural context, an individual's control over achievement tasks and the outcomes that shape value assessments also impact the emotion of achievement (Pekrun and Stephens, 2009; Loderer et al., 2020). For instance, differences in classroom environments, teaching quality, classroom goal structures and assessment procedures between cultures can affect student achievement emotions (Pekrun, 2018). Cross-cultural research has revealed that student assessments and social environments may vary between countries representing different cultures, leading to disparities in achievement emotions (Frenzel et al., 2007; Loderer et al., 2020). Moreover, emotions may differ during the acculturation process, where the challenges posed by acculturation may inhibit the development of emotions, which, in turn, has implications for achievement (Raccanello et al., 2020). In terms of learning and achievement environments, minority immigrant students experience negative emotions more frequently than majority students for various reasons, such as low status and the difficulties of living between two cultures (Frenzel et al., 2007; Raccanello et al., 2019; Raccanello et al., 2020). A study by Frenzel et al. (2007) found that Chinese students reported higher levels of joy, pride, anxiety and embarrassment and lower levels of anger than German students in mathematics. Raccanello et al. (2020) compared the negative achievement emotions (NAEs) experienced by minority and majority primary school students in Italy and mathematics. The results indicated that minority students reported heightened emotions of anger, shame and boredom toward Italian, while majority students experienced more intense anxiety and embarrassment towards mathematics. While several cross-cultural comparative studies are available in the literature, research on the achievement emotion of minority or refugee students is often overlooked. Moreover, there is a paucity of information available regarding the experiences of Syrian students. Therefore, this study aimed to address the gap in the literature by exploring the sense of achievements experienced by Syrian children who are victims of war in Türkiye.

## Method

### Sample

In this survey design, data were collected from 357 Syrian children under temporary protection who attended primary and secondary schools in Kilis on the Syrian border of Türkiye (Table 1). Of the participants, 173 (49%) were girls and 180 (51%) were boys. The ages of the participants ranged from 9 to 14 years ($M$ = 11.63; standard deviation [SD] = 1.66). Of the participants, 136 (38%) were primary school students and 221 (62%) were secondary school students; most had medium academic achievement levels. According to the grades used in the Turkish education system, students reported their mid-semester grades for mathematics, Turkish and science courses from 1 (very low) to 5 (very high). In addition, all participants attended public schools in the same classes as their Turkish peers.

### Measurements and ethical procedures

At the study's outset, ethical approval was obtained from the Ethics Committee of Kilis 7 Aralık University (Approval No: 2024/03). In addition, official permission was granted by the Kilis Provincial Directorate of National Education. Before data collection, school administrators, teachers and students were informed about the purpose and procedures of the study. Parents were notified through a written explanatory note prepared by the researchers. The note and parental consent form were sent home with the students, and parents who agreed for their children to participate signed and returned the forms to the teachers, who subsequently delivered

**Table 1.** Sociodemographic information for the sample

| Variables | % | *N* |
|---|---|---|
| **Gender** | | |
| Girl | 49.0 | 173 |
| Boy | 51.0 | 180 |
| **Age (years)** | | |
| 9 | 12.3 | 44 |
| 10 | 20.2 | 72 |
| 11 | 12.3 | 44 |
| 12 | 19.0 | 68 |
| 13 | 19.9 | 71 |
| 14 | 16.2 | 58 |
| **School level** | | |
| Primary school | 38.1 | 136 |
| Secondary school | 62.9 | 221 |
| **Parent (mother) education level** | | |
| Not going to school | 14.6 | 52 |
| Primary school | 13.7 | 49 |
| Secondary school | 34.7 | 124 |
| High school | 20.4 | 73 |
| University | 16.5 | 59 |
| **Parent (father) education level** | | |
| Not going to school | 9.5 | 34 |
| Primary school | 17.4 | 62 |
| Secondary school | 33.3 | 119 |
| High school | 18.2 | 65 |
| University | 21.6 | 77 |
| **The year in Türkiye** | | |
| 2 years or less | 3 | 0,8 |
| 3–5 years | 20 | 5,6 |
| 6–8 years | 74 | 20,7 |
| 9–11 years | 195 | 54,6 |
| 12 years and above | 65 | 18,2 |

them to the researchers. Written assent was also obtained from the participating children. They were informed that participation was voluntary and that they could withdraw or stop completing the questionnaire at any time without any negative consequences.

After the consent process was completed, a questionnaire including the Achievement Emotion Scale items and demographic questions was administered to the students. Those who agreed to participate completed the questionnaire in ~30 min. All data were collected through self-reports in face-to-face settings, and participants were assured that their responses would remain confidential and be used solely for research purposes.

**Achievement Emotion Scale:** This consists of items and was employed by researchers in accordance with control–value theory (Pekrun, 2006; Pekrun and Perry, 2014) as a means of assessment. The scale was developed based on data from a study conducted

using principal component analysis to determine the number of factors. In the second stage, an exploratory factor analysis was performed using a principal axis factor analysis with varimax rotation. In the third stage, factor charges were examined to assign items to the factors. Theoretical conformity was considered during this process. Items that did not meet the criterion of factor load |.40| or had a factor load below |.40| for at least two factors were not assigned to the factors. The feasibility of conducting exploratory factor analysis was established through Kaiser–Meyer–Olkin (KMO) = .98, and Bartlett's (p < .001) test analyses of the collected data. Principal component analysis in the factor analysis revealed a structure with an eigenvalue >1 for the scale, which explained 53.45% of the variance. When conducting exploratory factor analysis with varimax basic axis rotation, 35 items were found to load onto only one factor. The sum of the eigenvalues in the scale factors was 15.34, and the sum of the percentage of variance explained was 53.45. The factor loads of the items ranged from 0.44 to 0.81. Ultimately, the scale consists of two positive achievement emotions (PAEs; Joy and Pride) and five negative (Anger, Anxiety, Shame, Hopelessness and Boredom), all measured using a 5-point Likert structure. The scale's internal consistency was examined using Cronbach's alpha following factor analysis. The reliability of the subdimensions of the scale was found to be between 0.79 and 0.86 (Table 2).

### *Data analysis*

The Kolmogorov–Smirnov test was executed in the research study to determine whether the data followed a normal distribution. The findings indicated that the data exhibited a normal distribution in terms of both the items and the total attitude scores ($p > .05$). After conducting the Kolmogorov–Smirnov test, the researchers evaluated the distribution of the participants' achievement emotions using descriptive statistics, including the mean and SD. They performed an independent group $t$-test to analyze the differences in emotions based on gender and the participants' decision to continue their education at the upper level. Additionally, the researchers employed an analysis of variance test to examine the differences in emotions based on age, class and parental education level. They used the Scheffe post-hoc test to analyze the differences between the subgroups. Additionally, Pearson correlation analysis was used to examine the relationship between academic success and the sense of achievement based on the time spent in Turkey, and multiple linear regression analysis was used to investigate the effect of these variables on the feeling of accomplishment. In the data analysis conducted in this study, the significance level was set at $p < .05$.

### Results

The results in Table 2 reveal the general sentiments of achievement among the participants and the variations in these sentiments based on gender. The average PAE ($M = 3.33$) surpassed the average NAE ($M = 2.74$). Participants reported higher levels of enjoyment in PAEs ($M = 3.49$) compared to NAEs, such as anger ($M = 2.85$) and anxiety ($M = 2.81$). Students generally possess significantly more PAEs.

The emotional outcomes of the students were considerably differentiated by gender in terms of PAE ($t_{(1; 355)} = 2.09, p = .037$), NAE ($t_{(1; 355)} = -2.62, p = .009$), anger ($t_{(1; 355)} = -1.98, p = .049$), anxiety ($t_{(1; 355)} = -2.61, p = .009$), shame ($t_{(1; 355)} = -2.24, p = .026$) and

**Table 2.** Achievement emotion for girls, boys and the entire sample

| Achievement emotion | General | | | Girl (n = 173) | | Boy (N = 180) | | | |
|---|---|---|---|---|---|---|---|---|---|
| | α | M | SD | M | SD | M | SD | t | p |
| **PAE** | .87 | 3.33 | 0.87 | 3.43 | 0.88 | 3.24 | 0.85 | 2.09 | .037* |
| Enjoyment | .85 | 3.49 | 1.01 | 3.59 | 0.97 | 3.40 | 1.04 | 1.83 | .067 |
| Pride | .81 | 3.17 | 1.04 | 3.26 | 1.12 | 3.08 | 0.97 | 1.69 | .092 |
| **NAE** | .83 | 2.74 | 0.80 | 2.62 | 0.84 | 2.84 | 0.76 | −2.62 | .009* |
| Anger | .79 | 2.85 | 0.94 | 2.51 | 1.01 | 2.72 | 0.97 | −1.98 | .049* |
| Anxiety | .81 | 2.81 | 0.99 | 2.71 | 0.96 | 2.96 | 0.89 | −2.61 | .009* |
| Shame | .86 | 2.80 | 0.99 | 2.69 | 1.01 | 2.92 | 0.97 | −2.24 | .026* |
| Hopelessness | .82 | 2.62 | 1.01 | 2.66 | 1.03 | 2.91 | 0.93 | −2.32 | .021* |
| Boredom | .80 | 2.62 | 1.00 | 2.54 | 1.03 | 2.71 | 1.00 | −1.56 | .122 |

Abbreviations: NAE, negative achievement emotion; PAE, positive achievement emotion.

**Table 3.** Variation in achievement emotion by age

| Age | 9 years old (N = 44) | | 10 years old (N = 72) | | 11 years old (N = 44) | | 12 years old (N = 68) | | 13 years old (N = 71) | | 14 years old (N = 58) | | | |
|---|---|---|---|---|---|---|---|---|---|---|---|---|---|---|
| | M | SD | M | SD | M | SD | M | SD | M | SD | M | SD | F | p |
| **PAE** | 3.35 | 1.02 | 3.77 | 0.87 | 3.10 | 0.77 | 3.33 | 0.78 | 3.23 | 0.79 | 3.07 | 0.84 | 5.97 | .000* |
| Enjoyment | 3.37 | 1.17 | 3.86 | 1.04 | 3.18 | 1.00 | 3.57 | 0.90 | 3.43 | 0.92 | 3.30 | 0.98 | 3.55 | .004* |
| Pride | 3.33 | 1.02 | 3.67 | 1.01 | 3.01 | 0.92 | 3.08 | 1.05 | 3.02 | 0.98 | 2.83 | 1.05 | 5.75 | .000* |
| **NAE** | 2.65 | 0.77 | 2.61 | 0.89 | 2.82 | 0.84 | 2.86 | 0.79 | 2.88 | 0.78 | 2.60 | 0.68 | 1.68 | .139 |
| Anger | 2.38 | 0.90 | 2.45 | 1.10 | 2.85 | 1.10 | 2.79 | 1.01 | 2.73 | 0.91 | 2.52 | 0.86 | 2.12 | .062 |
| Anxiety | 2.86 | 0.96 | 2.80 | 1.03 | 2.72 | 0.90 | 3.03 | 0.90 | 3.02 | 0.85 | 2.57 | 0.92 | 2.25 | .049* |
| Shame | 2.90 | 1.01 | 2.72 | 1.03 | 2.89 | 1.07 | 2.85 | 0.93 | 2.91 | 0.99 | 2.66 | 0.93 | 0.69 | .631 |
| Hopelessness | 2.69 | 0.90 | 2.68 | 1.07 | 2.98 | 1.01 | 2.84 | 0.96 | 2.95 | 1.01 | 2.66 | 0.94 | 1.22 | .301 |
| Boredom | 2.42 | 0.93 | 2.39 | 1.06 | 2.68 | 1.04 | 2.77 | 1.06 | 2.81 | 1.05 | 2.62 | 0.84 | 1.90 | .093 |

*p<0.01.

hopelessness ($t_{(1; 355)}$ = −2.32, $p$ = .021). Specifically, girl students exhibited higher levels of PAE, whereas boy students experienced significantly greater levels of NAE, anger, anxiety, shame and hopelessness. These findings suggest that female students experience more PAEs, whereas male students experience more negative ones.

Table 3 demonstrates the alterations in the participants' achievement sensations based on age. As per age, PAE ($F_{(5; 351)}$ = 5.97, $p$ = .000), enjoyment ($F_{(5; 351)}$ = 3.55, $p$ = .004), pride ($F_{(5; 351)}$ = 5.75, $p$ = .000) and anxiety ($F_{(5; 351)}$ = 2.25, $p$ = .049) differed significantly. While the PAE and emotion of pride of the students in the 10-year-old group were higher than those in the 11-, 12-, 13- and 14-year age groups, the sense of pride was higher in the 11- and 14-year age groups. The 10-year-old Syrian students experienced more PAEs, such as joy and pride, than older students.

Table 4 shows the change in participants' achievement emotions according to their grade level. Significant differences in PAE by grade level ($F_{(4; 351)}$ = 3.70, $p$ = .006), pride ($F_{(4; 351)}$ = 5.92, $p$ = .000), NAE ($F_{(4; 351)}$ = 2.75, $p$ = .028), shame ($F_{(4; 351)}$ = 2.72, $p$ = .030) and boredom ($F_{(4; 351)}$ = 2.44, $p$ = .047) were evident for their emotions. PAE was founded among Syrian students in the fourth grade rather than in the sixth grade, and their sense of pride was higher than that among students in the sixth and eighth grades. Conversely, the

sense of shame was greater among sixth- and eighth-graders. Moreover, the sense of boredom was greater among seventh-grade students than among fourth-grade students.

Table 5 presents the alterations in the participants' sense of achievement based on their inclination to continue their education. A substantial difference was found in the experience of boredom ($t_{(1; 355)}$ = −2.32, $p$ = .021) between those who desired to continue their education and those who did not. Notably, boredom was significantly higher among students who harbored NAEs and were disinclined to pursue further education.

Table 6 depicts the variations in participants' achievement emotions based on parental education. Regarding the educational background of the mothers, significant differences in terms of pride ($F_{(4; 351)}$ = 5.92, $p$ = .000) and shame ($F_{(4; 351)}$ = 2.72, $p$ = .030) were observed in the emotions experienced by the participants. Similarly, regarding paternal education, PAE ($F_{(4; 351)}$ = 2.51, $p$ = .042), pride ($F_{(4; 351)}$ = 2.86, $p$ = .023) and shame ($F_{(4; 351)}$ = 2.55, $p$ = .039) also displayed significant variations in the emotions experienced by the participants. Notably, the sense of pride among Syrian students whose fathers were university graduates was higher than that among students whose fathers were secondary school graduates. This suggests that students whose fathers were university graduates tended to feel greater pride.

**Table 4.** Change in achievement emotion by grade level

| Grade level | Grade 4 (*N* = 136) | | Grade 6 (*N* = 65) | | Grade 7 (*N* = 119) | | Grade 8 (*N* = 32) | | *F* | *p* |
|---|---|---|---|---|---|---|---|---|---|---|
| | *M* | SD | *M* | SD | *M* | SD | *M* | SD | | |
| **PAE** | 3.54 | 0.94 | 3.13 | 0.78 | 3.25 | 0.80 | 3.19 | 0.87 | 3.70 | .006* |
| Enjoyment | 3.58 | 1.11 | 3.30 | 0.94 | 3.47 | 0.94 | 3.60 | 0.95 | 1.35 | .251 |
| Pride | 3.49 | 1.01 | 2.97 | 1.02 | 3.04 | 0.97 | 2.78 | 1.18 | 5.92 | .000* |
| **NAE** | 2.63 | 0.84 | 2.90 | 0.74 | 2.85 | 0.77 | 2.48 | 0.81 | 2.75 | .028* |
| Anger | 2.48 | 1.06 | 2.85 | 0.94 | 2.72 | 0.99 | 2.38 | 0.77 | 2.33 | .056 |
| Anxiety | 2.74 | 1.01 | 3.02 | 0.83 | 2.94 | 0.89 | 2.56 | 0.98 | 2.27 | .061 |
| Shame | 2.77 | 1.04 | 3.06 | 0.87 | 2.86 | 0.96 | 2.45 | 1.06 | 2.72 | .030* |
| Hopelessness | 2.69 | 1.00 | 2.97 | 0.91 | 2.90 | 1.00 | 2.54 | 0.98 | 1.77 | .134 |
| Boredom | 2.47 | 1.00 | 2.62 | 1.00 | 2.85 | 1.04 | 2.49 | 0.92 | 2.44 | .047* |

*$p<0.01$.

**Table 5.** Change in the achievement emotion according to the desire to continue education

| Willingness to continue education | Yes (*N* = 326) | | No (*N* = 31) | | *t* | *p* |
|---|---|---|---|---|---|---|
| | *M* | SD | *M* | SD | | |
| **PAE** | 3.35 | 0.86 | 3.06 | 0.91 | 1.81 | .071 |
| Enjoyment | 3.51 | 1.00 | 3.18 | 1.14 | 1.76 | .079 |
| Pride | 3.19 | 1.04 | 2.94 | 1.03 | 1.31 | .192 |
| **NAE** | 2.72 | 0.82 | 2.94 | 0.63 | −1.41 | .159 |
| Anger | 2.60 | 1.00 | 2.86 | 0.95 | −1.38 | .167 |
| Anxiety | 2.83 | 0.95 | 3.08 | 0.74 | −1.42 | .155 |
| Shame | 2.81 | 1.00 | 2.80 | 0.88 | 0.08 | .937 |
| Hopelessness | 2.79 | 0.99 | 2.92 | 0.95 | −0.69 | .489 |
| Boredom | 2.59 | 1.01 | 3.03 | 1.04 | −2.32 | .021* |

*$p<0.01$.

**Table 6.** Changes in Syrian students' achievement emotion according to parental education

| Maternal education | Didn't go to school (*N* = 52) | | Primary school (*N* = 49) | | Secondary school (*N* = 124) | | High school (*N* = 73) | | University (*N* = 59) | | *F* | *p* |
|---|---|---|---|---|---|---|---|---|---|---|---|---|
| | *M* | SD | *M* | SD | *M* | SD | *M* | SD | *M* | SD | | |
| **PAE** | 3.25 | 0.96 | 3.53 | 0.83 | 3.21 | 0.80 | 3.29 | 0.83 | 3.52 | 0.97 | 2.16 | .073 |
| Enjoyment | 3.35 | 1.14 | 3.62 | 0.98 | 3.43 | 0.95 | 3.52 | 0.98 | 3.58 | 1.11 | 0.67 | .614 |
| Pride | 3.15 | 1.00 | 3.45 | 0.96 | 2.99 | 1.02 | 3.06 | 1.04 | 3.46 | 1.10 | 3.28 | .012* |
| **NAE** | 2.79 | 0.76 | 2.79 | 0.84 | 2.85 | 0.77 | 2.53 | 0.82 | 2.68 | 0.84 | 2.03 | .090 |
| Anger | 2.58 | 0.94 | 2.67 | 0.96 | 2.79 | 1.00 | 2.39 | 0.91 | 2.54 | 1.12 | 2.02 | .092 |
| Anxiety | 2.93 | 0.90 | 2.94 | 1.01 | 2.89 | 0.93 | 2.63 | 0.92 | 2.89 | 0.94 | 1.27 | .282 |
| Shame | 2.94 | 0.99 | 2.93 | 0.98 | 2.93 | 0.92 | 2.54 | 1.04 | 2.70 | 1.04 | 2.49 | .043* |
| Hopelessness | 2.90 | 0.92 | 2.83 | 1.07 | 2.88 | 0.96 | 2.59 | 0.99 | 2.78 | 1.04 | 1.17 | .325 |
| Boredom | 2.59 | 1.04 | 2.60 | 0.93 | 2.77 | 1.01 | 2.50 | 1.09 | 2.52 | 0.98 | 1.06 | .377 |
| **Father education** | Not out of school (*N* = 34) | | Primary school (*N* = 62) | | Secondary school (*N* = 119) | | High school (*N* = 65) | | University (*N* = 77) | | *F* | *p* |
| | *M* | SD | *M* | SD | *M* | SD | *M* | SD | *M* | SD | | |
| **PAE** | 3.07 | 0.92 | 3.47 | 0.82 | 3.24 | 0.78 | 3.27 | 0.93 | 3.52 | 0.92 | 2.51 | .042* |
| Enjoyment | 3.23 | 1.03 | 3.63 | 0.97 | 3.48 | 1.00 | 3.35 | 1.01 | 3.61 | 1.06 | 1.43 | .224 |

(*Continued*)

**Table 6.** (*Continued*)

| Father education | Not out of school (N = 34) | | Primary school (N = 62) | | Secondary school (N = 119) | | High school (N = 65) | | University (N = 77) | | F | p |
|---|---|---|---|---|---|---|---|---|---|---|---|---|
| | M | SD | M | SD | M | SD | M | SD | M | SD | | |
| Pride | 2.92 | 1.14 | 3.31 | 0.95 | 3.00 | 1.02 | 3.19 | 1.07 | 3.42 | 1.02 | 2.86 | .023* |
| **NAE** | 2.93 | 0.77 | 2.92 | 0.80 | 2.75 | 0.80 | 2.63 | 0.79 | 2.59 | 0.82 | 2.21 | .067 |
| Anger | 2.84 | 1.07 | 2.72 | 0.93 | 2.68 | 0.94 | 2.58 | 0.92 | 2.39 | 1.13 | 1.72 | .145 |
| Anxiety | 3.15 | 0.90 | 3.01 | 1.03 | 2.81 | 0.90 | 2.72 | 0.91 | 2.76 | 0.92 | 1.86 | .118 |
| Shame | 3.05 | 1.02 | 3.05 | 0.94 | 2.82 | 1.00 | 2.57 | 0.90 | 2.71 | 1.05 | 2.55 | .039* |
| Hopelessness | 2.93 | 1.05 | 3.02 | 1.02 | 2.77 | 0.96 | 2.71 | 0.98 | 2.69 | 0.97 | 1.30 | .269 |
| Boredom | 2.71 | 1.00 | 2.80 | 0.95 | 2.68 | 1.06 | 2.59 | 0.99 | 2.40 | 1.00 | 1.51 | .199 |

*$p<0.01$.

**Table 7.** Correlation coefficients related to their achievement emotion, course achievements and time spent in Türkiye

| Variables | 1 | 2 | 3 | 4 | 5 | 6 | 7 | 8 | 9 | 10 | 11 | 12 | 13 |
|---|---|---|---|---|---|---|---|---|---|---|---|---|---|
| 1. PAE | – | | | | | | | | | | | | |
| 2. Enjoyment | .84* | – | | | | | | | | | | | |
| 3. Pride | .85* | .43* | – | | | | | | | | | | |
| 4. NAE | −.13* | −.18* | −.03 | – | | | | | | | | | |
| 5. Anger | −.17* | −.26* | −.04 | .82* | – | | | | | | | | |
| 6. Anxiety | .00 | −.05 | −.06 | .83* | .62* | – | | | | | | | |
| 7. Shame | −.02 | −.06 | −.02 | .83* | .55* | .69 | – | | | | | | |
| 8. Hopelessness | −.10* | −.12* | −.04 | .84* | .60* | .61* | .68* | - | | | | | |
| 9. Boredom | −.23* | −.25* | −.14* | .76* | .56* | .48* | 49* | 52* | - | | | | |
| 10. Math achievement | .28* | .24* | .23* | −.27* | −.23* | −.16* | −.17* | −.21* | −.30* | - | | | |
| 11. Turkish achievement | .27* | .21* | .25* | −.26* | −.27* | −.14* | −.13* | −.21* | −.30* | .58* | – | | |
| 12. Science achievement | .27* | .23* | .22* | −.27* | −.29* | −.13* | −.15* | −.21* | −.31* | .66* | .70* | – | |
| 13. Year of living in Türkiye | −.17* | −.16* | −.13* | .06 | .02 | .03 | .02 | .05 | .13* | −.18* | −.21* | −.25* | - |

*$p<0.01$.

Table 7 displays the correlation coefficients between the participants' achievement emotion and course achievement (Turkish-Mathematics-Science) levels and the time they spent in Türkiye. The correlation coefficients indicate positive correlations between Turkish ($r = .27$), mathematics ($r = .28$) and science achievements ($r = .27$), as well as PAE and its factors and negative correlations between NAE and its characteristics. Additionally, it was found that there are significant negative relationships between the year of living and PAE in Türkiye.

Table 8 shows the results of the multiple linear regression analysis conducted to determine participants' sense of achievement based on their academic performance (Turkish-Mathematics-Science) and the length of time spent in Turkey. The study revealed that academic achievement (Turkish-Mathematics-Science) and the length of time spent in Turkey had a low but significant positive relationship with positive achievement motivation ($R = .330$; $R^2 = .109$; $p < .001$) and a low but significant negative relationship with negative achievement motivation ($R = .308$; $R^2 = .095$; $p < .001$). Accordingly, academic achievement and time spent in Turkey explain 12% of the total variance in positive and 10% in negative achievement feelings. This indicates that other variables can explain the remaining portion of positive and negative achievement feelings. Furthermore, when examining the standardized regression coefficients, it is seen that mathematics achievement,

**Table 8.** Regression analysis of course s achievements and time spent in Türkiye as predictors of their achievement emotion

| Predictor variables | B | Std. Error | β | t | p |
|---|---|---|---|---|---|
| Positive achievement emotion (PAE) | | | | | |
| (Constant) | 3.086 | .273 | | 11.313 | .000 |
| Math achievement | .088 | .044 | .159 | 2.020 | .044* |
| Turkish achievement | .073 | .048 | .127 | 1.534 | .126 |
| Science achievement | .026 | .050 | .047 | .516 | .606 |
| Year of living in Türkiye | −.038 | .022 | −.103 | −1.716 | .087 |
| $N = 357$; $R = .330$; $R^2 = .109$; $F_{(4–269)} = 8.118$; $p < .001$ | | | | | |
| Negative achievement emotion (NAE) | | | | | |
| (Constant) | 3.351 | .254 | | 13.187 | .000 |
| Math achievement | −.069 | .041 | −.134 | −1.686 | .093 |
| Turkish achievement | −.060 | .045 | −.113 | −1.354 | .177 |
| Science achievement | −.055 | .046 | −.108 | −1.177 | .240 |
| Year of living in Türkiye | −.002 | .021 | −.007 | −.115 | .908 |
| $N = 357$; $R = .308$; $R^2 = .095$; $F_{(4–269)} = 6.951$; $p < .001$ | | | | | |

*$p<0.01$.

in particular, significantly predicts positive achievement feelings ($\beta$ = .159, $p$ < .05). According to the standardized regression coefficients ($\beta$), it can be said that the order of importance on the dependent variable is academic achievement (Mathematics, Turkish and Science) and the length of time spent in Turkey are significant predictors of the feeling of success. However, apart from mathematics achievement, these predictor variables do not substantially affect the sense of success.

## Discussion

This study investigated the emotional experiences of Syrian students studying in the same classes as their peers, focusing specifically on their examination performance based on gender, grade level, academic achievement, parents' educational status, desire to continue education and years spent living in Türkiye. Overall, it was found that students' PAEs were greater than their NAEs.

Previous research has identified gender, grade level and achievement as the variables most closely linked to the emotion of achievement. Numerous empirical studies on emotions have reported significant disparities in achievement levels among elementary and middle school students, based on grade level and achievement. Regarding sex, some studies have produced varying results.

In the present study, students' achievement emotions were assessed based on their gender, revealing that female students experienced significantly greater PAEs than male students. Conversely, boy students reported more NAEs than did girl students. Additionally, this study found that male students experienced significantly greater emotions of anger, anxiety, shame and hopelessness in their NAE than did female students. This difference between boy and girl students persisted despite their similar midterm grades.

In academic literature, mixed results have been reported in studies examining the impact of gender differences on achievement emotions. Some studies have found that girl students experience greater levels of anxiety than their boy counterparts (Bieg et al., 2015; Lohbeck et al., 2016) and report higher levels of hopelessness and shame (Frenzel et al., 2007), leading to increased NAEs. Much of this research has focused on mathematics. However, other studies have failed to detect a significant gender effect on achievement emotions. For instance, Harari et al. (2013) found no gender differences in math anxiety among US first-year students from high-minority schools. Lichtenfeld et al. (2012) reported similar emotional patterns among boy and girl students in an American elementary school sample. In a study conducted by Raccanello et al. (2019) to assess achievement emotion in both math and mother tongue domains with second- and fourth-graders, gender differences emerged only in the mother tongue domain, where boys reported feeling bored more than girls.

The current study may have revealed various reasons for girls reporting more PAEs than boys. Most of the studies that showed girls reported more NAEs than boys were conducted in mathematics. Girls in the current study may feel more achievements than boys overall, as they focus on their general achievement emotions. Another reason for this outcome could be that the students in the study were refugees, and boy and girl students may have been affected differently by this situation. Previous studies conducted with refugee students, such as those examining attitudes, motivation and adaptation, have shown favorable results for girl students.

In a study conducted by Halef (2021), it was reported that girls had a more positive attitude toward school than boys. Similarly, Acemioğlu et al. (2021) determined that girls had more positive attitudes toward science lessons than boys. In a study conducted by Akdeniz (2018), it was found that the level of adaptation (integration) of girls was significantly higher than that of boys. Mammadova (2022) found that girls were more willing to continue their education and achieve their life goals than boys. As these studies were conducted directly with Syrian students, it is noteworthy that similar gender patterns were observed in the current study. However, more empirical studies are needed in different schools and cultural contexts to evaluate the effect of gender on achievement emotions.

The study conducted by Raccanello et al. (2019) investigated the enjoyment, boredom and anxiety experienced by elementary students in mathematics classes and found that second-grade students reported higher enjoyment and lower levels of boredom and anxiety than fourth-grade students. Similarly, Vierhaus et al. (2016) conducted a study on the evaluation of achievement emotions between grades 5 and 7 and found that enjoyment levels were high and stable among students in grades 2–5, but decreased steadily among students in grades 5–7. Conversely, boredom started low in grade 2 and increased steadily over time. The literature suggests that a decrease in enjoyment and an increase in boredom are significant changes observed in students. According to Frenzel et al. (2007), the trajectories of enjoyment and boredom are reversed, with a decrease in enjoyment and an increase in boredom observed as students progress from early to middle adolescence. Researchers suggest that this may be due to a reduction in the cognitive activation and autonomy support provided by teachers, which negatively affects students' emotions of achievement and increases their experience of boredom.

Raccanello et al. (2020) reported a similar finding in their study with minority and majority elementary school students. The authors found that younger students experienced higher levels of positive emotions, whereas older students reported higher levels of negative emotions, such as anxiety. The researchers interpreted this finding as indicative of the students' growing inability to cope with the emotional consequences of their daily tasks. Additionally, the results of our study revealed a similar relationship pattern between achievement emotion and grade level among immigrant students.

Our findings regarding the relationship between achievement and achievement emotions align with previous research (Pekrun et al., 2002; Goetz et al., 2007). Most previous studies have focused on the association between enjoyment, anxiety, boredom and academic achievement. For example, Lichtenfeld et al. (2012) found that, among elementary school students, enjoyment in mathematics was positively correlated with achievement, whereas boredom and anxiety were negatively associated. Similarly, Van der Beek et al. (2017) conducted a study with classroom students. They found that high-achieving students experienced more enjoyment and less math anxiety than their average and low-achieving peers. Camacho-Morles et al. (2021) examined the relationships between joy, frustration, anger and boredom to better understand the impact of achievement emotions on achievement. The study confirmed a positive relationship between enjoyment and student achievement and a negative relationship with anger and boredom. Still, no direct relationship was found between disappointment and achievements. In a study of immigrants, Martin et al. (2024) reported that the second most important factor, after cultural factors, in explaining the emotions of high school students was pre-achievement. Achievement increased students' positive emotions at school and

decreased their negative emotions. These findings align with our study's findings, which revealed a positive relationship between achievement emotions and Syrian students' achievement.

Furthermore, the findings of our study, which reveal the significant impact of academic achievement and the length of time spent in Turkey on Syrian students' sense of accomplishment, support this conclusion. The association of positive feelings of success among Syrian students, particularly with mathematics achievement, indicates that cognitive competence plays a vital role in emotional well-being. However, the negative correlation between the length of time spent in Turkey and feelings of success suggests that the sociocultural and adaptation difficulties students encounter over time may negatively affect their feelings about success. Our findings indicate that only a limited part of feelings of success can be explained by academic achievement and length of stay in the host country. In contrast, the remaining part may be related to various individual, social or cultural factors.

Contrary to gender, grade level and achievement variables, no studies have examined the direct relationship between students' educational aspirations, parents' educational status and years of living in the country of migration with their achievement emotions. Findings related to these variables give us a different perspective on refugee students' achievement emotions. A positive correlation was found between educational continuation and achievement among Syrian students. Those who did not wish to continue their education reported feeling bored. Furthermore, parents' academic status has an impact on Syrian students' achievements. Specifically, mothers' educational attainment elicited the emotion of pride, whereas those with less or no education experienced the emotion of shame. Similarly, fathers' educational attainment was associated with more PAEs and a stronger sense of pride. Notably, the findings of this study indicate that a sense of pride is prominent when both parents have a high level of education.

As in the findings of our study, parental education emerged as a crucial factor in previous research on the relationship between students' parental education and emotion-based variables, such as educational aspirations and motivations (Mau and Bikos, 2000; Gil-Flores et al., 2011). Mammadova (2022) examined the factors affecting the educational aspirations and sense of school belonging of Syrian secondary and high school students. The results indicated that students whose parents had never attended school perceived themselves as more disadvantaged than those whose parents had attained university or higher education.

In addition, examining the relationship between years of residence in Türkiye and Syrian students' achievement emotions revealed a vital pattern: longer residence in the host country was associated with lower positive achievement and negative emotions. This finding may indicate that students who resided longer in the host country tend to report higher NAEs.

The migration process, as well as the attempt to reconcile the cultural differences between one's origin and the dominant culture, coping with estrangement from family and friends, acquiring a new language and facing discrimination, contribute to the psychological adjustment process, which can result in tension and distress (Suarez-Morales et al., 2007). This is especially true in learning and achievement environments, where minority immigrant children are likely to experience negative emotions. The challenges posed by acculturation may hinder the development of emotional competencies, affecting achievement emotions (Raccanello et al., 2019).

In the present study, it is possible that the increase in the number of years living in Türkiye was associated with a rise in NAEs among students. This may be linked to the sociocultural difficulties experienced by these students. Numerous studies have highlighted the problems faced by Syrian students in school environments (Emin, 2018; Arslan and Ergül, 2021; Tosun, 2021; Gökmen, 2020; Tanrıkulu, 2017). Eren's (2019). The study revealed that immigrant children experienced difficulties in their relationships with Turkish students and other peers and struggled to establish friendships. In research conducted by Börü and Boyacı (2016), it was noted that students who felt excluded in the school environment were more likely to form groups.

When reviewing studies conducted with Syrian students, it can be observed that the literature indicates that refugee students face challenges in adjusting to school because of cultural differences. As these negative experiences persist over time, prolonged exposure may be associated with stronger negative emotions toward achievement.

According to the structural equation model proposed by Martin et al. (2024) for the factors affecting the motivation and academic development of immigrant high school students, cultural demands and resources play a significant role in determining motivation and educational outcomes. Thus, according to researchers, cultural factors are critical in explaining students' school-related emotions. Our findings suggest that refugee students' achievement emotions may be influenced by factors such as gender, grade level, willingness to continue education beyond academic achievement, parents' educational background and years of living in the place of migration. These findings emphasize the significance of incorporating cultural determinants and variables, such as class, gender and achievement, when researching refugee students' emotions of achievement. Further empirical studies are needed to investigate the relationships between these variables and the minority/refugee population.

## Conclusion

The outcomes of our study advance the understanding of refugee students' achievement emotions, which have received scant attention in the literature. Our research sheds light on the positive and NAEs experienced by Syrian students and elucidates the connections between these emotions and several variables. Although our findings about gender, grade level and achievement variables are consistent with the existing literature on achievement emotion, they also introduce novel perspectives through variables such as the desire for education, the educational level of parents and the duration of residence in Türkiye. This information on how refugee students' emotional worlds differ based on various variables can guide researchers or educators who strive to facilitate students' adaptation processes in the context of their achievement emotions.

## Limitations

This study has several limitations that should be acknowledged. First, the research was conducted in a single city, Kilwith, with a unique demographic and sociocultural context due to its proximity to the Syrian border and its large refugee population. Therefore, the findings may not be fully generalizable to other regions of Türkiye or to different refugee-hosting contexts. Future research could benefit from including participants from multiple areas to explore whether similar patterns are observed across diverse settings.

Second, while the study provides valuable insights into the relationships between years of residence in Türkiye and Syrian

students' achievement emotions, the cross-sectional design limits the ability to infer causality. Longitudinal studies would allow a better understanding of how these emotional patterns develop and change over time.

Finally, the lack of qualitative data limits the depth of interpretation regarding students' emotional experiences. Incorporating qualitative approaches in future research could provide a more nuanced understanding of how refugee children experience and express their achievement emotions in educational settings.

**Open peer review.** To view the open peer review materials for this article, please visit http://doi.org/10.1017/gmh.2025.10089.

**Data availability statement.** The data generated and/or analyzed during the current study are not publicly available due to the ethics approval granted on the basis that only researchers involved in the study can access the de-identified data. The minimum retention period is 5 years from publication. Supporting documents are available upon request to the corresponding author.

**Author contributions.** SKC: Conceptualization, method, formal analysis, data curation and writing (original draft, reviewing and editing). FB and ES: Conceptualization, method and writing (reviewing). EK and EA: Conceptualization, method, formal analysis, data curation, supervision and writing (reviewing and editing). The author(s) read and approved the final manuscript.

**Competing interests.** The authors declare none.

**Ethical approval.** All procedures performed in studies involving human participants were in accordance with the ethical standards of the institutional and/or national research committee and with the 1964 Helsinki Declaration and its later amendments or comparable ethical standards. This study was approved by Kilis 7 Aralık University Social and Human Sciences Scientific Research and Publication Ethics Committee (confirmation number: 2024–2103). Informed consent has been waived due to its true nature.

**Informed consent.** No studies have been conducted on humans.

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
