## [Reviewer Report]

This study explores the achievement emotions of 357 Syrian refugee children attending primary and secondary schools in Turkey under temporary protection. The findings show that girls experience more positive achievement emotions (such as joy and pride), while boys report higher levels of negative emotions (such as anxiety, shame, and hopelessness). The study also finds that the longer students remain in Turkey, the more their positive achievement emotions decline. Variables such as age, grade level, parental education, and willingness to continue education are also shown to have significant effects on students' emotional experiences related to academic achievement.

The article addresses a highly relevant and under-researched topic with clear social and educational implications. It is well-grounded in theory, particularly Pekrun’s Control-Value Theory, and the quantitative analysis is methodologically sound. However, some limitations should be noted. The study is based on cross-sectional data, yet several claims are phrased in ways that suggest causality—this should be handled more cautiously. The presentation of the scale development process is overly technical and could be made more accessible. Furthermore, the absence of qualitative data limits the depth of interpretation regarding students’ emotional experiences. The sample is restricted to the city of Kilis, which may affect the generalizability of the findings.

Despite these limitations, the study makes a valuable contribution by shedding light on the emotional dimensions of refugee students’ academic lives. It offers useful insights for educators and policymakers and opens the door for further research, particularly using longitudinal or mixed-method approaches

---

## [Editor Report]

The manuscript examines achievement emotions among >350 Syrian refugee children in primary and secondary schools it Türkiye. The authors identify important gender differences and correlates of both positive and negative emotions, although the strength of these associations is usually quite weak. The paper addresses the critical and under-researched emotional dimension of refugee education. As one of the reviewers noted, this study is well grounded in existing theoretical frameworks and the analytic approach is appropriate for the research questions. I have provided some suggestions for the authors to consider alongside those of the reviewers.

I echo the reviewer’s observation that claims of causality should be removed from the paper. Given the cross-sectional design, statements implying that longer duration in Türkiye causes declines in positive emotions should be reframed and rephrased. The results are compelling and innovative, but they cannot establish temporal or causal relationships.

Second, the study is limited to one city (Kilis) and it is unclear how generalizable this setting is to other parts of Türkiye or to other contexts. The authors do a very nice job of introducing the broader context and rationale for focusing on refugee education in the intro. It may also be helpful to provide more information about the study setting and context, including how it is similar/different to other refugee-hosting regions of Türkiye or more broadly. This may also be relevant to elaborate on this point further in the limitations.

The study emphasizes statistical significance and pays limited attention to the magnitude of the estimates or the practical significance of the differences. I recommend that the authors incorporate interpretations of the magnitude of the associations into the results and discussion.

At times it seems that terms are used inconsistently, yet appear to be referring to the same concept (e.g., ‘achievement’ and ‘accomplishment’). Is this intentional? I suggest using consistent terminology to strengthen readability and clarity. 

In the discussion, the authors situate the study findings within the broader education literature. I recommend that the authors incorporate and engage more with the refugee education literature to contextualize the findings.

Lastly, please provide more clarity on the ethical approvals and procedures. In the methods, it states that there was an informed consent process that involved parents. Yet the additional information provided at the end of the manuscript suggests that there was a waiver of informed consent granted and this is not considered human subjects research. Please clarify and elaborate on this. Also, how were children approached and was there participation voluntary?

---

## [Reviewer Report]

I have carefully reviewed the revised version of the manuscript. The authors have addressed all of the previous comments and concerns comprehensively. The revisions have substantially improved the clarity, structure, and overall quality of the paper. In my opinion, the manuscript now meets the journal’s standards for publication. Therefore, I recommend that it be accepted for publication in its current form.

---

## [Editor Report]

Thank you for your thorough revision of the manuscript. All comments and suggestions have been addressed.